# On-surface synthesis of tailored organic platforms for single metal atoms

Amogh Kinikar [1,6], Xiushang Xu[2,6], Takatsugu Onishi[2],
Andres Ortega-Guerrero [1], Roland Widmer [1], Nicola Zema [3],
Conor Hogan [3], Luca Camilli [4], Luca Persichetti [4], Carlo A. Pignedoli [1],
Roman Fasel [1,5], Akimitsu Narita [2] ✉ & Marco Di Giovannantonio [3] ✉

Recent advances in nanomaterials have pushed the boundaries of nanoscale fabrication to the limit of single atoms, particularly in heterogeneous catalysis. Single atom catalysts, comprising minute amounts of transition metals dispersed on inert substrates, have emerged as prominent materials in this domain. However, overcoming the tendency of these single atoms to cluster beyond cryogenic temperatures and precisely arranging them on surfaces with desired local environments pose significant challenges. Employing organic templates for orchestrating and modulating the activity of single atoms holds promise. Here, we introduce an on-surface synthesis of a single atom platform wherein atoms are firmly anchored to specific coordination sites distributed along carbon-based polymers. These platforms exhibit atomic-level structural precision and stability, even at elevated temperatures, offering arrays of undercoordinated metal centers as model active sites for single-atom catalysis. We theoretically reveal the pronounced ability of these architectures to coordinate several gas molecules and experimentally visualize their interaction with CO and $CO_2$. Fine-tuning the structure and properties of the coordination sites offers unparalleled flexibility in tailoring functionalities, thus opening avenues for previously untapped potential in catalytic applications.

Catalysts play a critical role in modern society: from pharmaceuticals to metallurgy, improving the catalysts involved has led to better products that are more sustainably manufactured. Many catalysts are transition or rare earth metals[1], and one way to improve their efficiency, while simultaneously making them more sustainable, is to use their smallest functional units, leading to single-atom catalysts (SACs)[2,3]. This maximizes the utilization of the function-bearing entities, yielding an improved atom economy[4,5]. In SACs, the atoms are dispersed on and are supported by surfaces of host materials (Fig. 1a), typically ceramics such as zeolites[6], oxides[7], 2D materials such as

graphene[8–10], other metals such as copper[11–13], as well as metal-organic frameworks[14–16]. The isolated single atoms (SAs) exhibit properties distinct from conventional metal clusters and nanoparticles, and closely resemble the reactive centers in solution or gas phase[17]. As heterogeneous catalysis reaches this atomic frontier, the exact chemical environment of the single atoms defines their catalytic properties[18–20]. This leads to several challenges[3]: attaining a high density of individual atoms without the formation of metal clusters, ensuring that the majority of these atoms are in a desirable chemical state on the surface, and developing adaptable platforms compatible with various metals.

[1]Empa, Swiss Federal Laboratories for Materials Science and Technology, nanotech@surfaces Laboratory, Dübendorf, Switzerland. [2]Organic and Carbon Nanomaterials Unit, Okinawa Institute of Science and Technology Graduate University, Okinawa, Japan. [3]CNR – Istituto di Struttura della Materia (CNR-ISM), Roma, Italy. [4]Dipartimento di Fisica, Università di Roma "Tor Vergata", Roma, Italy. [5]Department of Chemistry, Biochemistry and Pharmaceutical Sciences, University of Bern, Bern, Switzerland. [6]These authors contributed equally: Amogh Kinikar, Xiushang Xu. ✉e-mail: akimitsu.narita@oist.jp; marco.digiovannantonio@cnr.it

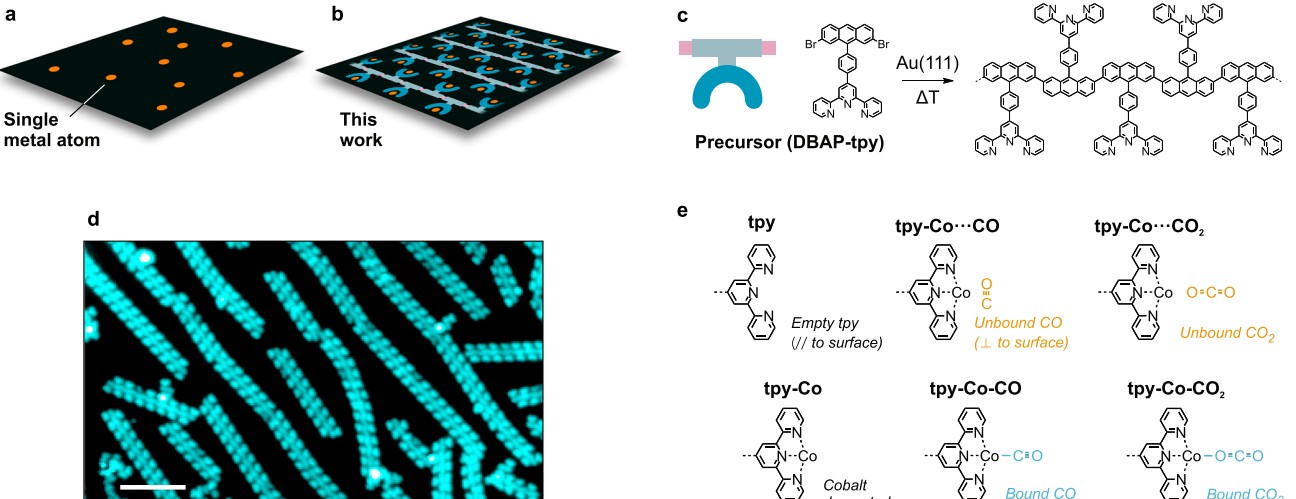

**Fig. 1 | Engineering single atom catalysts. a** Sketch of surface-adsorbed single atoms (orange dots). **b** Illustration of the SAP where the organic part is represented by 1D polymers adsorbed on a surface, and the single metal atoms decorate the coordination sites (U-shaped motifs). **c** The use of a DBAP-tpy molecular precursor enables the selective growth of 1D polymers via on-surface synthesis. **d** Large-scale experimental STM image of the SAP attained in this work. Scale bar: 10 nm. **e** Terpyridine functionalization motifs that are discussed in the text.

Consequently, achieving a high-density uniformly dispersed arrangement of single atoms, each with an atomically precise local environment, remains a longstanding objective. This goal also parallels the function of biological enzymes, where individual metal atoms are precisely coordinated within a protein structure, enabling a broad spectrum of catalytic actions[21,22].

Here, we introduce a surface-supported one-dimensional organic polymer with terpyridine (tpy) functional motifs[23] as side groups. These motifs can be chelated, post-synthesis, with a variety of metal atoms, making these polymers versatile atomically precise single-atom platforms (SAPs) (Fig. 1b). The atomic precision is achieved by triggering selective surface-catalyzed reactions of molecular precursors on single crystal Au(111) surfaces, a process often called on-surface synthesis (OSS)[24]. These organic polymers have several structural advantages due to the achieved structural precision, for instance, by having all the metal atoms in the same chemical environment, the elucidation of the structure-property relationship is simplified[25]. Compared to coordination polymers[26,27] and surface-confined metal-organic networks[28–32], which are also atomically precise and chemically active towards specific reactions, these polymers have metal coordination independent from polymerization. Not only does this enable the post-synthesis metal chelation, but it allows for asymmetrically coordinated metal atoms, which can be critical in improving the catalytic efficiency[21,33–35]. In addition, the $N_3$ open-site coordination achieved using the terpyridine motif also provides ample steric access to the coordinated atom. Lastly, although future catalysts for actual applications may rely on nanoparticle-based materials, having the SAPs on a clean, crystalline and atomically flat substrate such as Au(111) facilitates their fundamental characterization using powerful tools of surface science[36], particularly bond resolved scanning probe imaging that allows the direct visualization of active sites and bonding motifs.

We demonstrate the on-surface synthesis of an atomically precise SAP using a 4'-{4-(2,7-dibromoanthracen-9-yl)phenyl}−2,2':6',2''-terpyridine (DBAP-tpy) precursor (Fig. 1c) that undergoes dehalogenative aryl-aryl coupling on an Au(111) surface in ultrahigh vacuum (UHV) conditions. Post-synthesis, these polymers are coordinated with cobalt atoms. The resulting material is characterized by scanning tunneling microscopy (STM, example large-scale image in Fig. 1d) and non-contact atomic force microscopy (nc-AFM) with functionalized tips, complemented with density functional theory (DFT) simulations (performed with AiiDAlab based applications)[37]. DFT calculations

reveal that several gas molecules can be coordinated to these active sites and provide a rationalization of such behavior via in-depth analysis of the molecular orbital energetics and spatial localization. This theoretical investigation is complemented by the experimental visualization of CO and $CO_2$ binding configurations (Fig. 1e), providing fundamental insights into the potential of this material for single-atom reactivity studies.

## Results

The atomically precise SAP was synthesized through a three-step process. First, the DBAP-tpy precursor molecules were deposited on a clean Au(111) surface held at 200 °C under UHV conditions. This temperature promotes the debrominative homo-coupling yielding 1D polymers (Fig. 1c)[38]. High-resolution imaging of one of these polymer chains reveals a polyanthracene backbone with phenylene-terpyridine side moieties (Fig. 2a, b), as expected. Due to the precursor design and surface confinement, the steric hindrance operated by the tpy groups guarantees high selectivity in the covalent coupling, and only linear chains are formed[39,40]. While the tpy units appear as a Y-shaped feature in STM, nc-AFM with a CO-functionalized tip reveals their intramolecular structure as consisting of three planar rings. The central pyridine ring is slightly pulled up by the neighboring phenylene, which is rotated compared to the surface plane due to the hydrogen repulsion with the adjacent anthracene. Such rotation induces a periodic height modulation in the polyanthracene backbone, as highlighted in Fig. 2b. The nc-AFM image of a tpy unit (Fig. 2c) matches perfectly with the simulated one (Fig. 2d) obtained from the DFT-optimized geometry of the structure in Fig. 2e, indicating that the peripheral nitrogen atoms are pointing towards the polymer backbone. We also notice that the tpy units are slightly tilted towards the polymer long axis, probably due to the effect of the rotated phenylene rings and to a better match with the underlying surface lattice achieved in this configuration.

A sample with medium/high polymer coverage features chemisorbed bromine atoms that coexist with the polymer chains and stabilize them in islands (Supplementary Fig. 1a). If cobalt atoms are deposited on the surface at this stage, a cobalt-bromine complex is formed at the terpyridine (tpy-Co-Br, Supplementary Figs. 1b and 2). To achieve the targeted active sites featuring only cobalt as the coordinated atom, the second step of the platform preparation consists in the removal of bromine atoms. If this is attempted by annealing the surface at higher temperatures, some intramolecular transformations

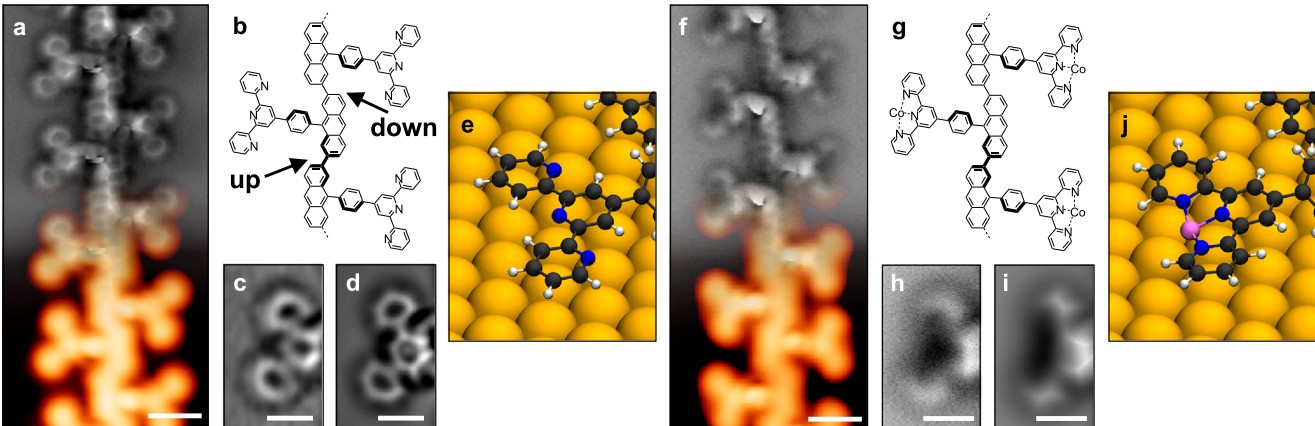

**Fig. 2 | Atomically precise SAP. a, f** Experimental STM (orange color scale, $I_t = 100$ pA, $V_b = -0.3$ V (**a**), $I_t = 50$ pA, $V_b = -0.1$ V (**f**)) and nc-AFM (gray color scale, $\Delta z = +$ 170 pm (**a**), $\Delta z = +190$ pm (**f**)) images of the 1D polymer obtained from the DBAP-tpy precursor on Au(111) via the OSS approach, before (**a**) and after (**f**) decoration with cobalt atoms. **b, g** Chemical structure of the obtained polymers. Bold segments indicate higher parts (i.e., further from the surface plane). **c, h** Zoom-in

experimental nc-AFM images ($\Delta z = +150$ pm (**c**), $\Delta z = +200$ pm (**h**)) of a tpy (**c**) and a tpy-Co (**h**) unit. **d, i** Simulated nc-AFM images of a tpy (**d**) and a tpy-Co (**i**) unit obtained from the structures in (**e**) and (**j**), respectively. **e, j** Zoom-in of the DFT-optimized geometries of a polymer segment on Au(111), without (**e**) and with (**j**) cobalt atoms coordinated to the tpy sites. Scale bars: 1 nm (**a, f**), 0.5 nm (**c, d, h, i**).

occur in the polymer before complete bromine desorption. Hence, we used an alternative, well-established procedure, which exploits atomic hydrogen[41]. This treatment (see Methods for the experimental details) successfully removes all bromine atoms from the surface and leaves the polymer chains (nearly) unaltered (Supplementary Figs. 1c and 3).

The third procedural step involves the controlled dosing of cobalt atoms, executed with the sample held at 150 °C to enhance surface diffusion and prevent cobalt clustering into islands. We meticulously calibrated the cobalt surface coverage to achieve a precise 1:1 ratio between tpy and Co. Subsequently, the sample underwent annealing at 200 °C to enhance order and cleanliness post the aforementioned steps. Upon this procedure, we found no indications of Co incorporation into the Au surface. As a result, the Y-shaped features observed by STM appear now less dark and less sharp in their center (Fig. 2f) as compared to the pristine polymers (Fig. 2a). nc-AFM reveals that the tpy moieties are no longer planar, as their constituent rings appear tilted down at the center of the unit (Fig. 2f, h). This is consistent with the presence of a cobalt atom coordinated to the tpy and closer in height to the surface plane, as evidenced by the good match with the simulated nc-AFM image (Fig. 2i) of a tpy-Co unit (Fig. 2j) and in agreement with a previous study using iron-coordinated tpy on a silver surface[42]. The distal pyridine rings are now pointing towards the complex center to maximize metal-ligand interaction. The tilt of the tpy-Co units towards the polymer long axis is still present. DFT-optimized geometries indicate that cobalt atoms occupy either hollow or bridge sites on the Au(111) surface, with local lifting of nearby gold atoms from the first atomic layer.

We affirm the successful on-surface synthesis of the targeted SAP, with cobalt loading that amounts to 10.9 wt%, or 1.7 at%. Samples with medium molecular coverage facilitated the growth of the SAP with quasi-evenly spaced chains adsorbed predominantly on the fcc regions of the Au(111) $22 \times \sqrt{3}$ (herringbone) reconstruction (Supplementary Fig. 1d). In such samples, the average density of active sites (tpy-Co) is approximately 0.15 per nm$^2$, corresponding to one every 6.6 nm$^2$. The DFT-calculated binding energy of cobalt atoms within the tpy pockets is 3.8 eV. Given that the sample underwent a temperature of 200 °C during preparation, this substantial binding energy anticipates the high stability of the active sites.

The well-defined structures obtained through our synthetic protocol feature cobalt atoms coordinated in an open N$_3$ environment. We computed the binding energies of several gas molecules (CO$_2$, CO, O$_2$, and H$_2$) at these active sites, as well as at a reference Co-porphyrin

complex, with all geometries optimized on a Au(111) surface via DFT. This analysis shows that – with the exception of CO$_2$, which interacts only weakly with both types of active sites – the tpy-Co centers exhibit stronger binding to the investigated gases than the Co-porphyrin (Fig. 3a). The enhanced binding strength at the tpy-Co sites can be advantageous for catalyzing reactions requiring selective stabilization of intermediates, such as CO in the CO$_2$ conversion to methanol and methane or to C$_{2+}$ products with two or more carbon atoms. To elucidate the origin of this increased reactivity, we analyzed the gas-phase molecular orbitals of the two model systems (Fig. 3b–g). These calculations reveal that the open coordination environment in the tpy-Co complex aligns the Co $d$-orbitals closely with the frontier molecular orbitals. Projected density of states (PDOS) analysis indicates that the frontier states are more localized on the cobalt center in the tpy-Co than in the Co-porphyrin. Deconvolution of the PDOS shows that the energy position of the 3$d$ orbitals of cobalt are significantly different in the two cases, with the $d_{xy}$, $d_{yz}$, $d_{xz}$ orbitals of the tpy-Co being closer to the frontier molecular orbitals than in the Co-porphyrin case (Supplementary Fig. 4). Moreover, in gas phase the two systems differ by their spin multiplicity (quartet for the tpy-Co and doublet for the Co-porphyrin) and by their formal charge state, with the tpy-Co deviating from the formal 2 + charge of the cobalt in Co-porphyrin (see the natural bond analysis in Supplementary Table 1). Thus, the combination of an open metal site and cobalt-centered frontier orbitals (some of which with Rydberg-like shape, like the HOMO−1 in Fig. 3d) emerges as a crucial electronic design, enhancing the potential for catalysis by facilitating bond activation and electron transfer processes. We emphasize that the electronic properties of these molecular systems may change upon adsorption onto the Au(111) surface. To provide a flavor of these modifications, we performed DFT single-point calculations of a tpy-Co in presence of a gold cluster, which revealed a distortion and symmetry breaking of the molecular orbitals (Supplementary Fig. 5). This broken symmetry is expected to promote more efficient electron transfer and stronger interactions with reactants, ultimately offering a spatial orbital distribution that is highly favorable for catalytic activity[43].

Cobalt, a pivotal metal in industrial processes and catalysis, plays a crucial role in diverse reactions, such as Fischer-Tropsch synthesis[44], oxidation of methane[45], steam reforming of ethanol[46] and methane[47], and is prohibitively expensive. The innovative SAP presented here comprises a minimal amount of this valuable material – yet our protocol is also applicable to other precious metals – and presents an ideal

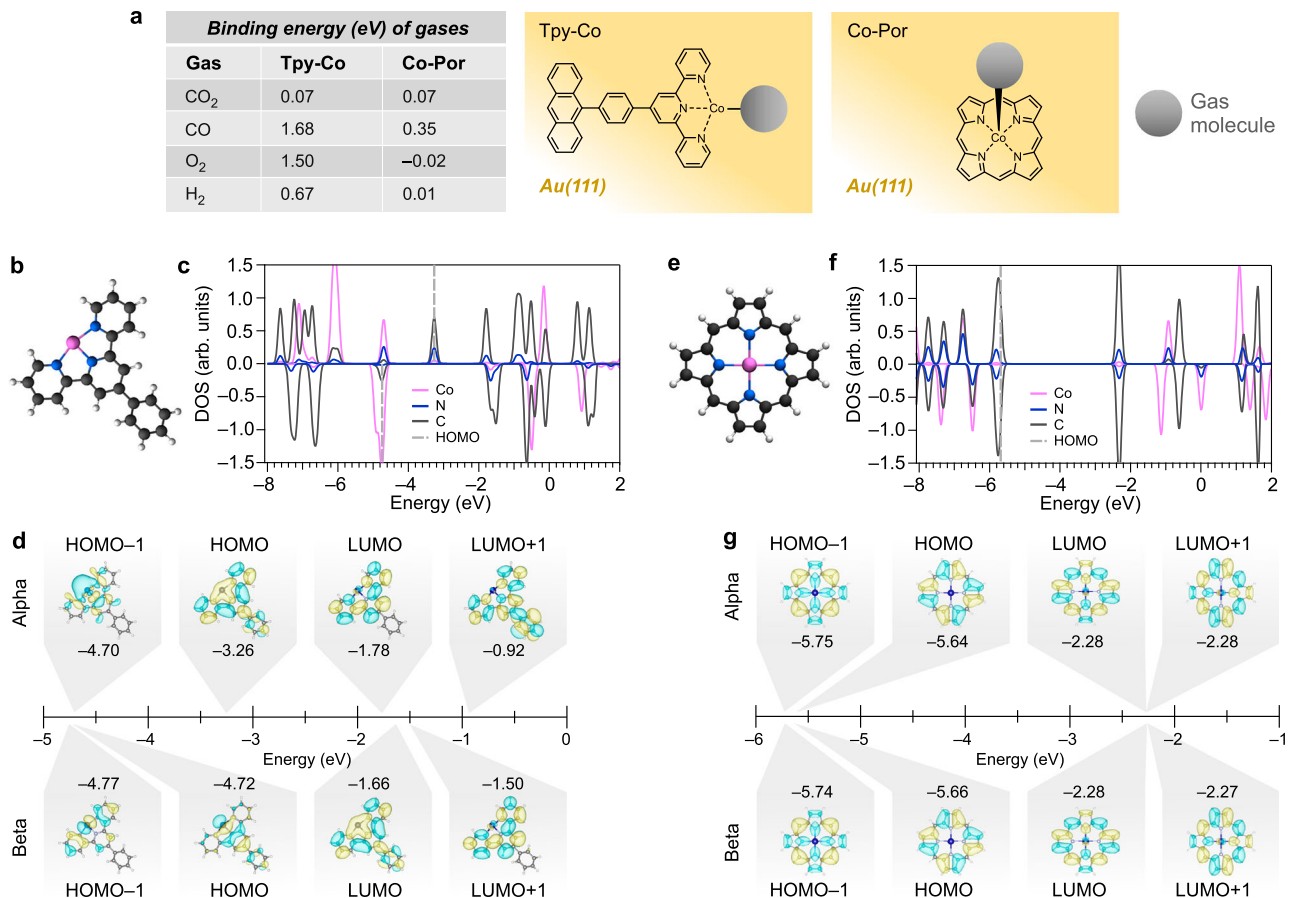

**Fig. 3 | Theoretical insights into the reactivity of the tpy-Co sites. a** Binding energies of several gases to the cobalt site of the tpy-Co system and of a reference Co-porphyrin. Optimized structures on A(111) obtained by DFT with periodic boundary conditions are sketched in the middle and right panels. The PBE exchange correlation functional is used with vdW corrections as proposed by Grimme (see "Methods"). These values were obtained by calculating the energy difference between the structure with the gas molecule interacting with the cobalt sites and adsorbed on the bare Au(111) surface. Multiple adsorption geometries were considered for the gas molecules, and the lowest energy were taken into account. **b** DFT-optimized gas phase molecular structure of a model tpy-Co system, which represents the lateral extensions of the 1D polymers synthesized herein. **c** PDOS for the tpy-Co, where positive (negative) values in the $y$-axis correspond to the spin-up (spin-down) channel. The highest occupied molecular orbital (HOMO) levels are indicated by the gray dashed lines. **d** Energy diagram of the frontier molecular orbitals of the tpy-Co. **e** DFT-optimized gas-phase molecular structure of a Co-porphyrin. **f** PDOS of the Co-porphyrin. **g** Energy diagram of the frontier molecular orbitals of the Co-porphyrin system. Isovalues for the probability densities in d,g are 0.012 bohr⁻³. The PBE0 functional with the def2-TZVP basis set was used in the gas-phase calculations.

opportunity to explore the performance of low cobalt doses under gas exposure. Based on the theoretical results described above, we focus on CO and $CO_2$ as model systems, also considering their central role in the mentioned industrial processes and $CO_2$ reduction.

We exposed the SAP to CO at a partial pressure of $3 \times 10^{-8}$ mbar for 90 s, with the Au(111) sample maintained at 9.0 K. This low dose typically results in a very low surface density of physisorbed CO molecules, used to functionalize the STM tip for improved imaging. The resulting STM image (Fig. 4a) reveals variations in tpy-Co sites, some appearing empty, while others exhibit dim or bright dots. Concurrent nc-AFM imaging clarifies these features (Fig. 4b). Empty units correspond to the tpy-Co sites described above. Dim dots in STM align with bright circular protrusions in nc-AFM (yellow arrows in Fig. 4b), resembling CO molecules physisorbed on the Au(111) surface and trapped between the polymer chains (white arrows in Fig. 4b). Hence, these units feature CO molecules standing perpendicular to the surface and stabilized by the cobalt atom without direct bonding (tpy-Co···CO in Fig. 1e). Less than 1% of units (statistics over more than 200 units) exhibit a bright center in STM (green circle in Fig. 4a). The corresponding nc-AFM image displays two faint features extending from the tpy (green circle in Fig. 4b), representing a CO molecule nearly parallel to the surface and covalently bound to the cobalt atom, i.e., tpy-Co-CO (Figs. 1e and 4c). The simulated nc-AFM image (Fig. 4e) of a DFT-optimized tpy-Co-

CO unit (Fig. 4f) perfectly aligns with the experimental image (Fig. 4d), confirming our assignment and consistent with a similar study on isolated molecules containing tpy-Fe moieties on Ag(111)[48]. We observed no appreciable structural changes with respect to the molecular adsorption site on the Au(111) surface.

Elevating the CO dose ($5 \times 10^{-8}$ mbar for 15 min) followed by a wobblestick annealing (WA, see Methods) for 15 min results in a sample, where 89% of the active sites (statistics over more than 800 units) exhibit bound CO molecules, specifically in the form of tpy-Co-CO (Fig. 4g and Supplementary Fig. 1e). First, the distinctive appearance of tpy-Co-CO sites in STM imaging demonstrates the potential for detecting CO at low temperatures, obviating the need for nc-AFM imaging. Then, it is noteworthy that the CO binds to the active sites via a post-adsorption diffusion mechanism. In fact, the tpy-Co-CO sites were only 1% before this treatment – with most CO adsorbed on the bare Au(111) or adjacent to the polymer chains – highlighting the key advantage of the tpy-based sites with open structure in contrast to closed pores or macrocycles. Complete CO desorption from the Au(111) surface and the tpy-Co sites is achieved after annealing the sample to room temperature, preserving the SAP for cyclic reuse without harm.

When exposing the SAP to $CO_2$ ($5 \times 10^{-8}$ mbar for 15 min) with the sample held at 30 K, the resulting surface exhibited chains with different occupation of the tpy-Co sites. Most of them show large, round

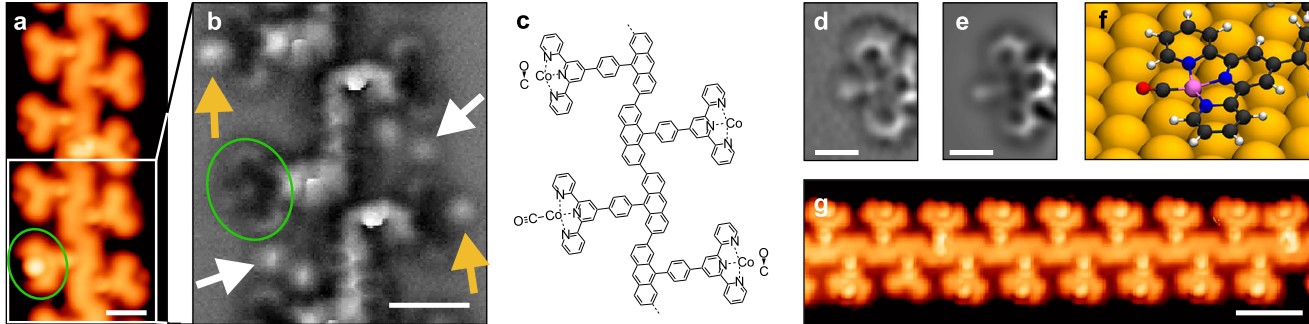

**Fig. 4 | Trapping of CO. a** Experimental STM image ($I_t = 70$ pA, $V_b = -0.1$ V) of a polymer segment where the tpy-Co sites are either empty, with bound CO, or with unbound CO. **b** nc-AFM image ($\Delta z = +220$ pm) of the region highlighted by the white box in (**a**). The round protrusions in between tpy units (white arrows) are physisorbed CO molecules stabilized next to the polymer backbone, and have a similar appearance to the CO molecules sitting next to the tpy-Co sites (tpy-Co···CO, yellow arrows). The green circles (**a**, **b**) highlight a tpy-Co-CO unit, where CO is bound to the active site. **c** Molecular structure of the segment and active site occupations in (**b**). **d** Experimental nc-AFM image ($\Delta z = +140$ pm) of a tpy-Co-CO site. **e** Simulated nc-AFM image of the tpy-Co-CO unit obtained from the structure in (**f**). **f** Zoom-in of the DFT-optimized geometry of a polymer segment on Au(111), with tpy-Co-CO units. **g** Experimental STM image ($I_t = 100$ pA, $V_b = -0.1$ V) of a polymer segment with nearly all units featuring bound CO molecules. Scale bars: 1 nm (**a**, **b**), 0.5 nm (**d**, **e**), 2 nm (**g**).

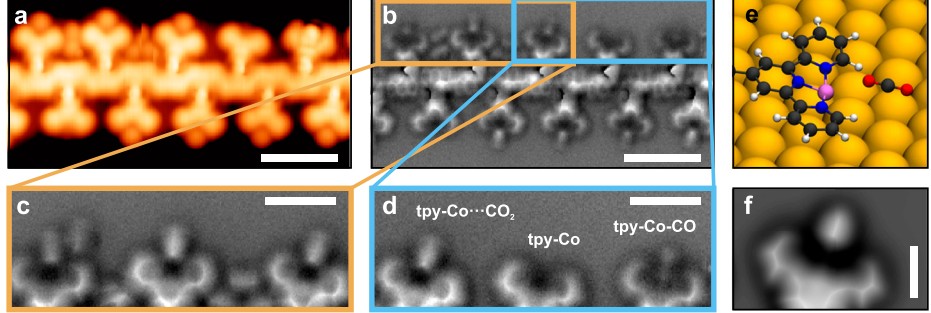

**Fig. 5 | Trapping of CO₂. a, b** Experimental STM ($I_t = 100$ pA, $V_b = -0.1$ V) and nc-AFM ($\Delta z = +240$ pm) images of a polymer segment where the tpy-Co sites are either empty or with $CO_2$ sitting next to them. A tpy-Co-CO site is also visible (top right). **c, d** Zoom-in nc-AFM images ($\Delta z = +220$ pm) of the areas highlighted by the yellow and cyan boxes in (**b**). **e** DFT-optimized geometry of a polymer segment on Au(111), with tpy-Co···$CO_2$. **f** Simulated nc-AFM image of the tpy-Co···$CO_2$ unit obtained from the structure in (**e**). Scale bars: 2 nm (**a**, **b**), 1 nm (**c**, **d**), 0.5 nm (**f**).

features (Fig. 5a), which appear as elongated, rod-like objects in nc-AFM (Fig. 5b, c). Those next to the tpy-Co are slightly nonplanar. The simulated nc-AFM image of a DFT-optimized tpy-Co···$CO_2$ unit (Fig. 5e, f) shows an excellent match with the experimental images. The rod-like feature of the $CO_2$ is fully reproduced, including its nonplanar appearance due to a weak interaction with the Co. The calculated distance between cobalt and the nearest oxygen is 2.81 Å. This value, significantly larger than the distance between cobalt and carbon in the case of tpy-Co-CO units (1.74 Å), suggests that the $CO_2$ is not bound to the cobalt, but weakly stabilized next to it (tpy-Co···$CO_2$, see Fig. 1e and Supplementary Fig. 9). DFT-optimized geometries of this structure with varying Co-$CO_2$ distances (Supplementary Fig. 9) offer further insights into the possible binding of $CO_2$ to Co, which, despite being accompanied by a significant change in morphology, does not produce a significant energy gain – in agreement with our theoretical results (Fig. 3a). The coexistence of "empty" tpy-Co sites with those decorated with CO and $CO_2$ (Fig. 5d), offers a direct comparison of the possible functionalization motifs and highlights the strength of our characterization down to the atomic level. Remarkably, we have further investigated the sample after stepwise annealing and observed the diminishing of tpy-Co···$CO_2$ sites in favor of new species (Supplementary Fig. 10) whose morphology (studied via STM and nc-AFM) is compatible with either tpy-Co-CO (discussed previously) or tpy-Co-$CO_2$ (Supplementary Fig. 9a, b). A definitive determination of the fate of $CO_2$ upon annealing would require dedicated investigations using complementary spectroscopic techniques – such as x-ray photoelectron spectroscopy (XPS) and temperature-programmed desorption

(TPD) – which are beyond the scope of the present study. Nevertheless, we provide direct visualization of key molecular species and their interactions with the active sites described herein, establishing a robust platform for future studies on single-atom reactivity.

In conclusion, we have developed a strategy for the synthesis of model SAPs based on organic polymers functionalized with terpyridine side groups that can be selectively chelated post-synthesis, achieving all-equivalent and regularly spaced active sites. These structures differ from isolated molecular systems on surfaces that present a broad distribution of site occupancy and poor spacing control. The resulting asymmetric $N_3$ coordination environment around the metal centers yields open-site structures that are highly promising for studies of single-atom reactivity. Using high-resolution scanning probe microscopy with functionalized tips, we achieved atomic-level characterization of these unique active sites. Furthermore, we computed the binding energies of several gases (CO, $O_2$, $H_2$, and $CO_2$) at the tpy-Co centers and compared them to those at a metalloporphyrin reference. Our results demonstrate that the tpy-Co sites exhibit significantly stronger binding to CO, $O_2$, and $H_2$ – a combination of properties that is advantageous for catalytic reactions such as $CO_2$ conversion to methanol, methane and $C_{2+}$ products via Fischer-Tropsch processes. The enhanced reactivity of the tpy-Co active sites is attributed to their distinct electronic structure, characterized by frontier molecular orbitals that are more localized on the metal center than in traditional systems like porphyrins. This localization facilitates bond activation and electron transfer processes. In addition, we directly visualized the binding configurations of CO and $CO_2$ molecules at the active sites through scanning probe microscopy, providing

experimental confirmation of their adsorption behavior. Our work establishes a new route for the synthesis of single-atom platforms with an atomically defined environment that serves as a reference for understanding single-site reactivity and enables the isolation of specific chemical effects – complementary to more complex or industrially relevant systems. Given that the synthesis is based on well-established on-surface chemistry protocols, it can be readily extended to different molecular precursors, allowing systematic tuning of the structural and electronic properties of the resulting material[24,49–51]. In the future, we envisage that such atomically engineered platforms could be rationally designed to selectively catalyze targeted reactions, approaching the efficiency and specificity typically associated with biological enzymes.

## Methods

### General methods for precursor synthesis
All reactions working with air- or moisture-sensitive compounds were carried out under an argon atmosphere using standard Schlenk line techniques. All starting materials, reagents, and solvents were purchased from commercial sources and used as received unless otherwise noted. Anhydrous dimethylformamide (DMF), tetrahydrofuran (THF), and dichloromethane ($CH_2Cl_2$) were purified by a solvent purification system (GlassContour) prior to use. Thin-layer chromatography (TLC) was done on silica gel coated aluminum sheets with F254 indicator, and column chromatography separation was performed with silica gel (particle size 0.063–0.200 mm). Nuclear Magnetic Resonance (NMR) spectra were recorded in $CDCl_3$ using Bruker Avance Neo 400 and 500 MHz NMR spectrometers. Chemical shifts ($\delta$) were expressed in ppm relative to the residual of solvents ($CDCl_3$, $^1H$: 7.26 ppm, $^{13}C$: 76.00 ppm). Coupling constants ($J$) were recorded in Hertz. Abbreviations: $s$ = singlet, $d$ = doublet, $t$ = triplet, $m$ = multiplet. High-resolution mass spectra (HRMS) were recorded on a Bruker ultra-fleXtreme spectrometer by matrix-assisted laser desorption/ionization (MALDI) with tetracyanoquinodimethane (TCNQ) as the matrix, or on a Thermo Scientific LTQ-Orbitrap Mass Spectrometer by electrospray ionization (ESI).

### On-surface experiments
The on-surface synthesis experiments were performed under ultrahigh vacuum (UHV) conditions with base pressure below $2 \times 10^{-10}$ mbar. Au(111) substrates (MaTeck GmbH) were cleaned by repeated cycles of $Ar^+$ sputtering (1 keV) and annealing (460 °C). The precursor molecules were thermally evaporated onto the clean Au(111) surface held at 200 °C from quartz crucible heated at 170 °C (deposition rate of ~ 0.5 Å·min$^{-1}$). Atomic hydrogen was dosed onto the surface held at 150 °C by using a gas cracker (Mantis) operated at 70 W with a partial pressure of $2 \times 10^{-7}$ mbar of molecular hydrogen (Messer, purity 5.0) for 15 min. Cobalt atoms were sublimated onto the surface held at 150 °C from a rod (Alfa Aesar, purity 4.5) heated in a home-built e-beam metal evaporator, and the surface coverage was controlled using the STM and nc-AFM images as feedback. After cobalt deposition, the sample was annealed to 200 °C for 20 min. CO, $CO_2$, and Ar gases (Linde, purity 4.7, Pangas, purity 4.5, and Messer, purity 5.0, respectively) were dosed onto the cold sample in the STM stage with a partial pressure of $5 \times 10^{-8}$ mbar for 15 min. The sample temperature reached 9.5 K during the exposure with the shields of the STM cryostat kept open. In some cases, we performed an unusual annealing of the sample to prevent potential contamination from the STM stage that might occur in the case of standard resistive heating. Such wobblestick annealing (WA) consisted in extracting the cold sample from the STM stage with the (room temperature) pincer tool that is usually operated for transfers (namely, the wobblestick), and holding the sample on this pincer for a specific time. Afterwards, the sample is reinserted into the cold STM stage and cooled down to 4.7 K. While not allowing any temperature readout, this method guarantees the cleanest annealing conditions.

### STM and nc-AFM imaging
STM images were acquired with a low-temperature STM/nc-AFM (Scienta Omicron) operated at 4.7 K in constant-current mode using an etched tungsten tip. The scanning parameters are indicated in the captions, with bias voltages referred to the sample. nc-AFM measurements were performed at 4.7 K with a tungsten tip placed on a qPlus force sensor[52]. The tip was functionalized with a single CO molecule at the tip apex picked up from the previously CO-dosed surface[53]. The sensor was driven at its resonance frequency (27405 Hz) with a constant amplitude of 70 pm. The frequency shift from the resonance of the tuning fork was recorded in constant-height mode using Omicron Matrix electronics and HF2Li PLL by Zurich Instruments. The $\Delta z$ reported in the captions is positive (negative) when the tip-surface distance is increased (decreased) with respect to the STM setpoint at which the feedback loop is open ($I_t$ = 100 pA, $V_b$ = − 5 mV, with the tip located on an empty area of the Au(111) surface).

### Computational details
**On surface.** All calculations were performed with AiiDAlab[37] apps based on the DFT code CP2K[54]. The surface/adsorbate system was modeled within the repeated slab scheme, with a simulation cell containing up to 1500 atoms, of which 4 atomic layers of Au along the [111] direction and a layer of hydrogen atoms to passivate one side of the slab in order to suppress one of the two Au(111) surface states. 40 Å of vacuum was included in the simulation cell to decouple the system from its periodic replicas in the direction perpendicular to the surface. The size of the cell was $38.3 \times 51.1$ Å$^2$ corresponding to 260 Au(111) surface unit cells. The electronic states were expanded with a TZV2P Gaussian basis set[55] for C and H species and a DZVP basis set for Au and Co species. A cutoff of 600 Ry was used for the plane wave basis set. Norm-conserving Goedecker-Teter-Hutter pseudopotentials[56] were used to represent the frozen core electrons of the atoms. We used the PBE parameterization for the generalized gradient approximation of the exchange correlation functional[57]. To account for van der Waals interactions, we used the D3 scheme proposed by Grimme[58]. To obtain the equilibrium geometries, we kept the atomic positions of the bottom two layers of the slab fixed to the ideal bulk positions, and all other atoms were relaxed until forces were lower than 0.005 eV/Å. The Probe Particle model[59] was used to simulate nc-AFM images. The binding energy of gas molecules to the active sites described in Fig. 3a have been computed by comparing the DFT-optimized geometries of the molecular systems with the gas molecule either bound (or in proximity) to the active sites to the case where the gas molecules were adsorbed on the Au(111) surface, far from the active sites – multiple geometries have been tested and the most stable ones were used for the comparison.

**Gas phase.** The molecular orbitals and PDOS reported in Fig. 3b–g have been computed using geometries optimized in Gaussian 16[60] with the PBE0[61] exchange-correlation functional. The def2-TZVP basis set[62] was used for all elements. The tpy-Co (Co-porphyrin) system displayed a quartet (doublet) spin state as the lowest energy configuration in the gas phase.

From the optimized structure of tpy-Co on an Au(111) surface discussed in the main text, a representative model containing an Au$_4$ cluster and the tpy-Co molecule was defined (see its geometry in Supplementary Fig. 5a, b). In this case, pseudopotentials were employed for the Au atoms. A visualization of the resulting frontier molecular orbitals is reported in Supplementary Fig. 5c. The spin configuration of the system tpy-Co + Au$_4$ cluster was evaluated for two multiplicities of Co ($M$ = 2 and 4), with the doublet spin state identified as the lowest energy configuration (Supplementary Table 2). Geometry optimization calculations were performed on the tpy-Co molecule in the gas phase, excluding the Au$_4$ cluster, using both spin multiplicities as well (Supplementary Table 3 and Supplementary Figs. 6 and 7). In addition, molecular orbitals of Co-

porphyrin were computed for comparison with those of the tpy-Co complex at the gas-phase (Supplementary Fig. 8).

## Data availability

The data supporting the findings of this study are available within the paper and its Supplementary Information. The source data files are available via Materials Cloud at DOI: 10.24435/materialscloud:yp-bp. All data are available from the corresponding author upon request.

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

## Acknowledgements

We acknowledge the European Union, Next Generation EU, Mission 4, Component 1, CUP B53D23013760006 and the Italian Ministry of University and Research (MUR) for the PRIN 2022 (project No. 2022JW8LHZ, ATYPICAL (M.D.G.)), the Okinawa Institute of Science and Technology Graduate University, the CNR-JSPS (Italy-Japan) bilateral project 2023- 2024 (project No. JPJSBP120234004, ACCESS (M.D.G., A.N.)), the CNR Short Term Mobility program, the Swiss National Science Foundation under Grant No. 212875 (R.F.), the NCCR MARVEL funded by the Swiss National Science Foundation (205602, C.A.P.). Computational support from the Swiss Supercomputing Center (CSCS) under project ID s1267 (C.A.P.) and lp83 (C.A.P.) is gratefully acknowledged. We acknowledge PRACE for awarding access to the Fenix Infrastructure resources at CSCS, which are partially funded from the European Union's Horizon 2020 research and innovation program through the ICEI project under the grant agreement No. 800858 (C.A.P.). We are thankful to Lukas Rotach (Empa) and Massimiliano Rinaldi (CNR-ISM) for their excellent technical support during the experiments and gratefully acknowledge Dr. Sobi Asako (RIKEN) for fruitful discussions.

## Author contributions

Conceptualization: A.N. and M.D.G.; Funding acquisition: C.A.P., R.F., A.N., and M.D.G.; Investigation: A.K., X.X., T.O., A.O.G., R.W., C.H., and C.A.P.; Project administration: R.F., A.N., and M.D.G.; Resources: R.W., N.Z., C.H., C.A.P., R.F., A.N., and M.D.G.; Supervision: R.F., A.N., and M.D.G.; Visualization: A.K., A.O.G., C.A.P., and M.D.G.; Writing – original draft: A.K., L.P, L.C, C.A.P., R.F., A.N., and M.D.G.; Writing – review & editing: all authors.

## Competing interests

The authors declare no competing interests.
