## [Transparent Peer Review file · Nature Communications]

On-Surface Synthesis of Tailored Organic Platforms for Single Metal Atoms

Corresponding Author: Dr Marco Di Giovannantonio

Version 0:

Reviewer comments:

Reviewer #1

(Remarks to the Author)

The manuscript describes a method to synthesize metal-organic structures featuring cobalt atoms in asymmetric threefold-coordinated sites. These metal-organic structures were synthesized on Au(111), characterized by scanning probe microscopy (STM and ncAFM) and subsequently exposed to CO and CO₂ in ultrahigh vacuum conditions at cryogenic temperatures. The individual adsorbate molecules were imaged by STM and ncAFM, and their appearance was compared to simulated ncAFM images based on DFT-optimized models.

A crucial aspect of the work is the claim that the reactivity of the asymmetric cobalt sites studied in this work is very distinct from cobalt sites in other metal-organic structures. This is illustrated by DFT computations indicating that the binding strengths of adsorbed molecules (CO, CO₂, O₂, H₂) is much higher than in a chosen reference system (Co-porphyrin). Unfortunately, this point is currently not supported by the presented experimental data, as the experiments do not provide any information about binding strengths of the imaged adsorbate molecules, nor do they provide any comparison to reference systems (such as the Co-porphyrin, which was chosen as a reference in the DFT computations).

In the cover letter (responding to a previous review of the manuscript), the authors state that traditional spectroscopic methods used in model catalysis research cannot be applied for this system due to the inherently low surface coverage of the Co sites. This limits the potential impact of the work, but the authors argue that this limitation can be outweighed by the atomic-scale definition of their system, allowing direct comparison to DFT models that can lead to fundamental insights inaccessible by ensemble-averaging methods. Such argumentation is fair, but the manuscript currently does not present the promised fundamental insights.

To provide such insights (that I would deem necessary for publication in Nature Communications), the following steps can be suggested:

1. Calibrate the temperature of the very vaguely defined “wobble-stick annealing” and estimate the temperature at which the adsorbed molecules desorb from the cobalt sites. This way, it will be possible to provide experimental values that can be compared to the DFT results presented in Figure 3a.
2. Provide some experimental reference of an alternative Co-site (ideally the Co-porphyrin that was analyzed by DFT in Fig. 3a), to provide experimental evidence that the binding strengths of adsorbed molecules are indeed dramatically different from the asymmetric cobalt sites.
3. Analyze the DFT data: why are the binding strengths so dramatically different between the Co-tpy and Co-porphyrin sites? Do the variations in binding strengths correlate to any electronic parameters of the Co-sites? (formal charge state, spin multiplicity, d-band position, hybridization of specific d-orbitals...?)
4. It is not clear why the authors chose to plot the PDOS and energy diagrams of a coordinatively saturated bis(tpy)-Co system in Figure 3e-g. The reactivity of bis(tpy)-Co was not addressed at all, thus I don't see the point of analyzing its PDOS and relating it to reactivity. Moreover, the bis(tpy)-Co system does not seem to be a suitable reference, as it can be expected that its reactivity will be significantly reduced by steric hinderance (which may outweigh any differences stemming from the distinct electronic structure). Without any doubt, the electronic structure Co-porphyrin should be analyzed instead, and it should be clearly explained what parameters are responsible for the differences in reactivity.

To conclude, I see this manuscript as a nice surface science work. I recognize the potential of this (or similar) system to potentially serve as a suitable model for elucidating the fundamentals of "single-atom" site reactivity. Nevertheless, for publication in Nature Communications I would expect to see some of fundamental understanding that this system can bring – not only promises that it may be done in the future.

Without such deeper insights I can only recommend publication in a more specialized journal.

Version 1:

Reviewer comments:

Reviewer #2

(Remarks to the Author)

This work presents an original on-surface polymerization reaction of a terpyridine based ligand which is then used to coordinate cobalt atoms (tpy-Co). The Co-terpyridine polymer system is subsequently utilized to actuate a ligation between the Co centers and C=O molecules (tpy-Co-C=O). This chemistry is studied at a single molecule level employing UHV-STM and nc-AFM. The reactivity of the on surface tpy-Co system and to a reference, Co-porphyrin molecule, toward C=O are explored using theory. The results of this study are highly relevant to the current interest in developing single-atom catalysts. Overall, I found this work quite interesting and have the following questions and comments:

It is not clear to me what exactly is calculated and depicted in Figure 3. Are these slab calculations, periodic boundary calculations, or gas phase calculations? If the former, why bother with gas phase calculations? If the later, why not use the slab or PBC calculations which will more accurately represent the energies? It is well known that the substrate can have a dramatic effect on the axial binding by Cobalt II porphyrins.

Terpyridine is a labile molecule in solution or gas phase and the external N's can either be pointed to form a pocket as shown, or oriented in the opposite direction. What is causing them to all adopt the pocket configuration?

The chemistry reported here is essentially the same (Co replacing Fe) as in a previous communication (Nat. Comm. 2022, 13, 7407); reference 48 in the manuscript, mentioned in one sentence on page 6. The authors should discuss their results in the context of that work.

Point-by-point response

Reviewer 1

3. Analyze the DFT data: why are the binding strengths so dramatically different between the Co-tpy and Co-porphyrin sites? Do the variations in binding strengths correlate to any electronic parameters of the Co-sites? (formal charge state, spin multiplicity, d -band position, hybridization of specific d -orbitals...?)

Our reply: We are grateful to the reviewer for this comment, which gives us the opportunity to further elucidate the origin of the higher reactivity of our tpy-Co compared to traditional system with known electronic configurations, such as a Co-porphyrin. We have performed additional calculations to display the electronic properties of the Co-porphyrin, including a new PDOS graph and energy diagram of the molecular orbitals (new Fig. 3e-g in the main text). The PDOS comparison confirms that the frontier states are more localized on the cobalt center in the tpy-Co than in the Co-porphyrin. Deconvolution of the PDOS reveals that the energy positions of the cobalt $3d$ orbitals differ significantly between the two systems. In the tpy-Co complex, the d_{xy} , d_{yz} , and d_{xz} orbitals lie closer to the frontier molecular orbitals compared to the Co-porphyrin case (new Supplementary Fig. 4). In the Co-porphyrin, the frontier orbitals correspond to the well-known π molecular orbitals, where the two HOMOs are nearly degenerate (a_{1u} and a_{2u} under D_{4h} symmetry) and the two LUMOs are exactly degenerate (e_g under D_{4h}). Moreover, the two systems differ by their spin multiplicity (quartet for the tpy-Co and doublet for the Co-porphyrin in gas phase). Natural Bond Orbital (NBO) analysis comparing the natural electron configuration of the Co atom in both systems shows a similar overall configuration with a slight deviation of the tpy-Co from the known Co(II) complex with a formal charge of +2 and one unpaired electron of the Co-porphyrin. In particular, the comparison reveals that the Co atom in tpy-Co exhibits a higher d -orbital occupation (new Supplementary Table 1). These differences arise from the distinct coordination environment of tpy-Co, which is less coordinated and more open than that of Co-porphyrin, ultimately leading to increased chemical reactivity.

New Supplementary Fig. 4 | PDOS of Co $3d$ orbitals. **a.** PDOS of the $3d$ orbitals of cobalt in the tpy-Co compound. **b.** PDOS of the $3d$ orbitals of cobalt in the Co-porphyrin. The graphs elucidate the origin of the spectral features reported in Fig. 3c,f in the main text. Positive (negative) values in the y-axis correspond to the spin-up (spin-down) channel. Highest occupied molecular orbital (HOMO) levels are indicated by the grey dashed lines.

New Supplementary Table 1 | Summary of the Natural Population Analysis (NPA) from the Natural Bond Orbital (NBO) Analysis of the Co atom in tpy-Co and Co-porphyrin.

Natural electron configuration Co atom	tpy-Co	Co-porphyrin
Alpha	[core]4s(0.19)3d(4.84)4p(0.18)	[core]4s(0.14)3d(4.30)4p(0.20)
Beta	[core]4s(0.07)3d(3.03)4p(0.08)	[core]4s(0.10)3d(3.29)4p(0.19)
Total	[core]4s(0.26)3d(7.86)4p(0.26)4d(0.01)	[core]4s(0.25)3d(7.59)4p(0.38)4d(0.01)

The results are presented as the effective valence electron configuration, also referred to as the *Natural Electron Configuration*. The analysis is provided separately for the α and β spin densities, as well as for the overall total. The reported values show a higher *d* orbital occupancy for the tpy-Co.

Action: We have performed new gas-phase calculations for a Co-porphyrin system and included the results in Fig. 3e-g. Moreover, we have added to the SI the deconvolution of the PDOS into the *d* orbitals of cobalt for the two complexes (new Supplementary Fig. 4) and the NBO analysis for the description of the formal charge state (new Supplementary Table 1). Finally, we have carefully described the differences in the electronic properties of the tpy-Co and Co-porphyrin in light of the new computational results, offering a rationalization of the increased chemical reactivity of the newly developed active sites.

4. It is not clear why the authors chose to plot the PDOS and energy diagrams of a coordinatively saturated bis(tpy)-Co system in Figure 3e-g. The reactivity of bis(tpy)-Co was not addressed at all, thus I don't see the point of analyzing its PDOS and relating it to reactivity. Moreover, the bis(tpy)-Co system does not seem to be a suitable reference, as it can be expected that its reactivity will be significantly reduced by steric hinderance (which may outweigh any differences stemming from the distinct electronic structure). Without any doubt, the electronic structure Co-porphyrin should be analyzed instead, and it should be clearly explained what parameters are responsible for the differences in reactivity.

Our reply: We understand the reviewer's concern about our choice to display and discuss the electronic properties of a bis(tpy)-Co. With this choice, we wanted to highlight the difference between our tpy-Co system and a coordinatively saturated one. However, in line with the reviewer's request, we have now computed the electronic properties of a Co-porphyrin and discussed them in comparison to the tpy-Co. A detailed description is reported in our answer to the previous point.

Action: The theoretical results in Fig. 3e-g have been replaced with those of a Co-porphyrin. Moreover, we have discussed the differences in the electronic properties of this system in comparison to the tpy-Co and added in the SI a new figure (new Supplementary Fig. 4, showing a deconvolution of the PDOS into the *d* orbitals of cobalt) and a new table (new Supplementary Table 1, with the NBO analysis).

Point-by-point response

Reviewer 2

This work presents an original on-surface polymerization reaction of a terpyridine based ligand which is then used to coordinate cobalt atoms (tpy-Co). The Co-terpyridine polymer system is subsequently utilized to actuate a ligation between the Co centers and C=O molecules (tpy-Co-C=O). This chemistry is studied at a single molecule level employing UHV-STM and nc-AFM. The reactivity of the on surface tpy-Co system and to a reference, Co-porphyrin molecule, toward C=O are explored using theory. The results of this study are highly relevant to the current interest in developing single-atom catalysts.

Our reply: We appreciate the reviewer’s positive evaluation of our work and address their comments below.

Overall, I found this work quite interesting and have the following questions and comments:

It is not clear to me what exactly is calculated and depicted in Figure 3. Are these slab calculations, periodic boundary calculations, or gas phase calculations? If the former, why bother with gas phase calculations? If the later, why not use the slab or PBC calculations which will more accurately represent the energies? It is well known that the substrate can have a dramatic effect on the axial binding by Cobalt II porphyrins.

Our reply: We thank the reviewer for the opportunity to clarify the calculations presented in Fig. 3. The geometries shown in Fig. 3a correspond to DFT-optimized structures of the molecular units adsorbed on Au(111), obtained using periodic boundary conditions. In contrast, Figs. 3b-g illustrate the electronic properties of two compounds whose structures (indicated in Fig. 3b and Fig. 3e) were optimized via gas-phase DFT calculations. Gas-phase optimization was chosen to enable the use of more accurate functionals, which would be computationally prohibitive for the full molecule-substrate system. Although substrate effects are neglected under these conditions, this approach allowed us to identify distinct differences in the electronic properties of the two systems and to gain deeper insight into the higher reactivity of the tpy-Co structure relative to the reference Co-porphyrin complex.

Action: In the caption of Figure 3, we clarified that the calculations in panel a are “optimized structures on A(111) obtained by DFT with periodic boundary conditions”.

Terpyridine is a labile molecule in solution or gas phase and the external N's can either be pointed to form a pocket as shown, or oriented in the opposite direction. What is causing them to all adopt the pocket configuration?

Our reply: The reviewer is correct that the distal pyridine rings can rotate, causing the external nitrogen atoms to point either inward (toward the ligand center) or outward (toward the polyanthracene backbone). The rotation of similar rings in surface-adsorbed nanostructures has been previously investigated by us and was found to occur with only a modest energy input (J. Am. Chem. Soc. 2018, 140, 10, 3532–3536). Consistent with this, our nc-AFM measurements of the “empty” tpy ligands (Fig. 2a,c) closely match the expected appearance of structures with outward-oriented distal pyridine rings (Fig. 2d,e). In certain cases – such as after prolonged annealing, as in our

hydrogen exposure experiments – we observed a different contrast in the tpy structures, in agreement with the simulated nc-AFM image of the tpy ligand with inward-oriented distal pyridine rings (Supplementary Fig. 3). Upon cobalt coordination, the distal pyridine rings adopt the inward orientation, reflecting stabilization of the complex when all three rings participate in metal-ligand coordination (Nat. Commun. 2022, 13, 7407).

Action: In the main text of our revised manuscript, we clarified that after coordination of cobalt atoms “the distal pyridine rings are pointing towards the complex center to maximize metal-ligand interaction”.

The chemistry reported here is essentially the same (Co replacing Fe) as in a previous communication (Nat. Comm. 2022, 13, 7407); reference 48 in the manuscript, mentioned in one sentence on page 6. The authors should discuss their results in the context of that work.

Our reply: We thank the reviewer for their comment, which gives us the opportunity to clarify the context and novelty of our work. The central innovation of our study lies in the design and on-surface synthesis of a single-atom platform in which undercoordinated, open active sites are interconnected through a covalently bonded carbon-based nanostructure. This concept is fundamentally different from the cited work (Nat. Commun. 2022, 13, 7407), even though the chemical nature of the active sites is comparable (tpy-Co on Au(111) vs. tpy-Fe on Ag(111)). Our goal was not only to visualize active sites and their local environment upon gas adsorption, but to achieve such insights within an extended, well-defined platform that could serve as a prototype for studying and tuning single-atom reactivity. In this regard, we developed a nanomaterial featuring regularly spaced and chemically equivalent active sites – distinct from isolated molecules that exhibit a broad distribution of site functionalization and occupancy. Furthermore, the gases used to probe the platform’s response were CO and CO₂ in our study, whereas CO and C₂H₄ were employed in the cited work. Finally, by comparing the electronic properties of our system with reference metal-porphyrins, we provided a deeper understanding of the asymmetric coordination environment and its associated higher reactivity. In summary, although the active site chemistries share similarities, the two studies differ in their objectives, material design, and characterization strategies. Together, they address complementary aspects of the dynamic field of single-site molecular catalysts at the interface between homogeneous and heterogeneous catalysis.

Action: We have highlighted the context of our study in the final part of our revised manuscript, adding that we achieved “all-equivalent and regularly spaced active sites” and that “these structures differ from isolated molecular systems on surfaces that present a broad distribution of site occupancy and poor spacing control”.